# Long-Term Temporal Changes of Precipitation Quality in Slovak Mountain Forests

**Jozef Minďaš [1,2,*], Miriam Hanzelová [3], Jana Škvareninová [4], Jaroslav Škvarenina [3,*], Ján Ďurský [2] and Slávka Tóthová [5]**

[1] Institute of Ecology and Environmental Sciences, University of Central Europe, Kráľovská 11, 90901 Skalica, Slovakia

[2] Ecological & Forestry Research Agency EFRA, T. G. Masaryka 8041, 96053 Zvolen, Slovakia; jan.dursky@azet.sk

[3] Department of Natural Environment, Faculty of Forestry, Technical University in Zvolen, T. G. Masaryka 24, 96000 Zvolen, Slovakia; mirowka@gmail.com

[4] Department of Applied Ecology, Faculty of Ecology and Environmental Sciences, Technical University in Zvolen, T. G. Masaryka 24, 96000 Zvolen, Slovakia; skvareninova@tuzvo.sk

[5] National Forest Center, T. G. Masaryka 22, 960 01 Zvolen, Slovakia; tothova@nlcsk.org

* Correspondence: jozefmindassk@gmail.com (J.M.); skvarenina@tuzvo.sk (J.Š.); Tel.: +421 915 552 744 (J.M.); Tel.: +421-455-206-209 (J.Š.)

**Abstract:** The paper is focused on the evaluation of long-term changes in the chemical composition of precipitation in the mountain forests of Slovakia. Two stations with long-term measurements of precipitation quality were selected, namely the station of the EMEP (European Monitoring and Evaluation Programme) network Chopok (2008 m a.s.l.) and the station of the ICP Forests (International Co-operative Programme on Assessment and Monitoring of Air Pollution Effects on Forests) network Poľana-Hukavský grúň (850 m a.s.l.). All basic chemical components were analyzed, namely sulfur (S-SO4), nitrogen (N-NH4, N-NO3), and base cations (Ca, Mg, and K) contained in precipitation. The time changes of the individual components were statistically evaluated by the Mann–Kendall test and Kruskal–Wallis test. The results showed significant declining trends for almost all components, which can significantly affect element cycles in mountain forest ecosystems. The evaluated forty one-year period (1987 to 2018) is characterized by significant changes in the precipitation regime in Slovakia and the obtained results indicate possible directions in which the quantity and quality of precipitation in the mountainous areas of Slovakia will develop with ongoing climate change.

**Keywords:** precipitation quality; mountain forests; long-term changes; acid components; base cations

## 1. Introduction

The Slovak Republic is a party of the UN ECE (United Nations Economic Commission for Europe) Convention on Long-range Transboundary Air Pollution. The Implementing Protocols have been progressively adopted to this Convention which, among other things, has been designated by the parties to the Convention to reduce the anthropogenic emissions of pollutants involved in global environmental problems. The Global Environment Report in Europe [1], published by the European Environment Agency, indicates that emissions of acidifying substances have declined significantly since 1990, mainly in Central and Eastern Europe due to economic restructuring. The reduction in Western Europe is mainly related to changes in fuel use, desulfurization, and denitrification of combustion gases and the introduction of three-way catalytic converters in cars. Due to significant emission reductions

in most European ecosystems, there is no further acidification, but there are a number of risk areas, especially in Central Europe [2–4].

The need of evaluation of the extent and distribution of pollutants over Europe initiated "The co-operative programme for monitoring and evaluation of the long-range transmission of air pollutants in Europe" (unofficially "European Monitoring and Evaluation Programme" = EMEP). The main objective is to provide governments/decision makers with information of the deposition and concentration of air pollutants, as well as the quantity and the significance of the long-range transmission of air pollutants and their fluxes across boundaries [5,6].

The main source of SOx emissions are large point sources from combustion in energy and transformation industries (56%), but its share decreased by 4% compared to 2012 [7]. Within the last two decades a distinct decrease in emissions in Czech Republic, Slovakia, and Poland has been achieved due to improvement of heat and power production technology, implementation of measures aimed at reducing sulfur emissions from power stations that use brown coal, increased use of low-sulfur fuels in private households and emission control at national and European level [8,9]. Some studies show that precipitation containing a higher proportion of sulfates are more toxic than precipitation at the same pH level but containing a higher proportion of nitrates [10,11].

In Europe, nitrogen oxides (NOx-N) are emitted mainly from stationary combustion sources (power plants and industrial processes) and transportation sources (road, off-road, and ship traffic). NOx-N emissions increased in the 1980s due to increased traffic and started to decrease in the mid-1990s [2,11]. The road transport sector represents the largest source of NOx emissions, accounting for 39% of total EU-28 emissions in 2013 [1,11].

More than 95% of all NH3 emissions come from agriculture, livestock production, and animal waste management. NH3 emissions from the use of artificial nitrogen fertilizers are also significant. NH3 emissions from energy/industry and transport are less significant. NH3 emissions from industry mainly come from the production of nitric acid. NH3 emissions from transport come mainly from road transport. In terms of long-term development, the overall NH3 emissions decrease. The decline in NH3 emissions in the "Agriculture" sector is due to the combined effect of reduced livestock numbers across Europe (especially cattle), changes in the handling and management of organic manures and the abatement of nitrogenous fertilizers [1,12].

The main goal of the article was to analyze the significance of temporal changes of selected parameters of the chemical composition of rainwater in two mountain localities in Slovakia. The aim was to determine the statistical significance of the decrease/increase in the concentrations of selected chemical components in precipitation during the evaluated time period as well as to determine the significance of the differences between the two evaluated localities.

## 2. Materials and Methods

### 2.1. Site Description

Chopok is the SHMI (Slovak HydroMeteorological Institute) meteorological observatory located on the ridge of the Low Tatras Mts., at an altitude of 2008 m, (48°56′ N, 19°35′ E) (Figure 1). Measurements started in 1977. Since 1978 it has been part of the EMEP network and the GAW(Global Atmosphere Watch)/BAPMoN(Background Air Pollution Monitoring Network)/WMO(World Meteorological Organisation) network. Chopok is located in a cold climatic zone, C3—cold mountain (July mean temperature below 10 °C). The long-term mean annual precipitation total (1951–1980) is 1142 mm and of which 667 mm is the summer half-year. The mean annual temperature (1951–1980) is −1.2 °C, in the vegetation period (April–September) 3.6 °C. A more detailed climatic characteristic is given by the climate chart (Figure 2) [13].

Research plot Polana Hukavský grúň is located in the territory of the Polana biosphere reserve at an altitude of 850 m (48°38′ N, 19°29′ E) (Figure 1). The mean annual precipitation total (1951–1980) is 853 mm and of which 494 mm in the summer half-year. The mean annual temperature (1951–1980) is

5.8 °C, during the vegetation period (April–September) 11.9 °C. A more detailed climatic characteristic is given by the climate chart (Figure 2).

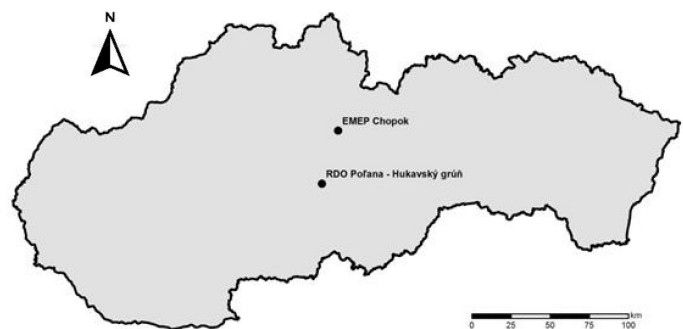

**Figure 1.** Location of the mountain monitoring stations EMEP (European Monitoring and Evaluation Programme) Chopok and RDO (Research Demonstration Object) Poľana.

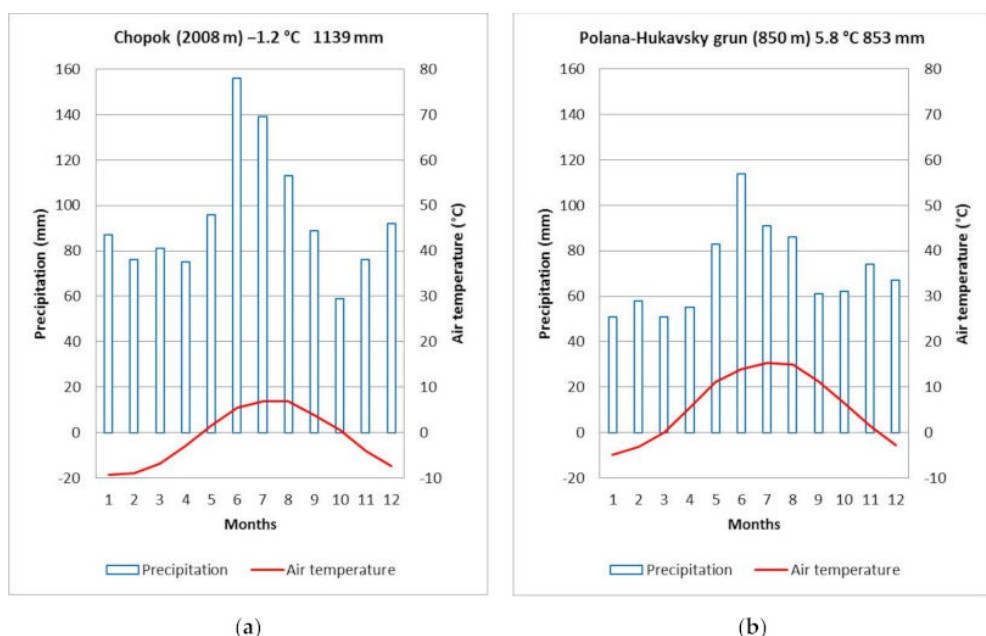

**Figure 2.** Climate diagrams for (**a**) Chopok and (**b**) Polana-Hukavsky grun.

## 2.2. Precipitation Quality Data

Source data for this article has been obtained for Chopok public data available in the EMEP database. For Polana, the data obtained from the Polana-Hukavsky Grúň Research and Demonstration Facility (VDO) was established by the Forest Research Institute in Zvolen in the spring of 1991. Since 1992, the basic components of precipitation quality have been monitored. It takes place in the weekly measurement and sampling cycle during the summer and the two-week cycle in the winter period is observed in 80% of cases.

The design of precipitation water sampling and the implementation of chemical analyzes of precipitation were performed in accordance with the EMEP manual [14] for the Chopok station and the ICP Forests manual [15] for the Polana-Hukavsky grun station. Chemical analyzes were performed in accredited laboratories of SHMI (Chopok) and Forest Research Institute in Zvolen (Polana) in accordance with the cited manuals. Both laboratories have been involved in international ring testing programs and related quality programs (quality assurance/quality control).

At the Chopok station, total precipitation was measured with a standard METRA rain gauge with wind protection with a collected area of 500 cm$^2$, samples for chemical analyzes were collected

into plastic PET containers with a collected area of 500 cm$^2$, in a wet-only version for summer period and bulk version for winter period. At the Polana site, precipitation totals were measured with a Hellman's rain gauge with a collected area of 200 cm$^2$, samples for chemical analyzes were collected into PET bulk collectors (3 pieces with an individual collected area of 200 cm$^2$). The precipitation quality measurements at Polana site have been changed due to organizational reasons and therefore the statistical analyzes have been performed up to 2014 year.

*2.3. Data Analysis and Statistical Methods*

The sulfates, nitrates, and ammonium have been recalculated according to the atomic weights to sulfur and nitrogen concentrations.

In the evaluation, individual statistical characteristics are evaluated for annual mean values at monitored stations. Annual weighted means of concentrations shall be assessed. For detecting and estimating trends of annual means have been used the non-parametric Mann–Kendall Test. It is the test of a monotonic increasing or decreasing trend. It is standard method when occur missing values and when data are not normally distributed. Sen's method can be used in cases where the trend can be assumed to be linear. It is an estimator used to quantify the magnitude of potential trends. Thus, Sen's slope is used to estimate the percent reduction in the concentration level while the Mann–Kendall test is used to indicate the significance level of the trend [5,6,16,17]. This test has been used for various studies on long-term environmental and economic impacts [18–21].

Decrease of concentration was calculated from differences of the first three and the last three years averages (depending on availability of measured data) [8]. The rate of decrease of the concentration of the analyzed element in precipitation was expressed on the basis of the regression coefficient "a" of the linear trend from the equation y = a.x + b.

The non-parametric Kruskal–Wallis test was performed by comparing the weighted averages of the stated values from the open areas between the stations Chopok and Poľana.

## 3. Results

*3.1. Precipitation*

In the monitored period 1978–2018, the annual precipitation total was recorded at the Chopok regional station, which varied in the range of 840–1590 mm, as pointed out in Figure 3. The maximum was recorded in 2014. From the statistical assessment by the Mann–Kendall test and the graphical representation of the values, it is clear that in the long run there is a slight increase in precipitation totals at a rate of 8.8 mm of atmospheric precipitation per year.

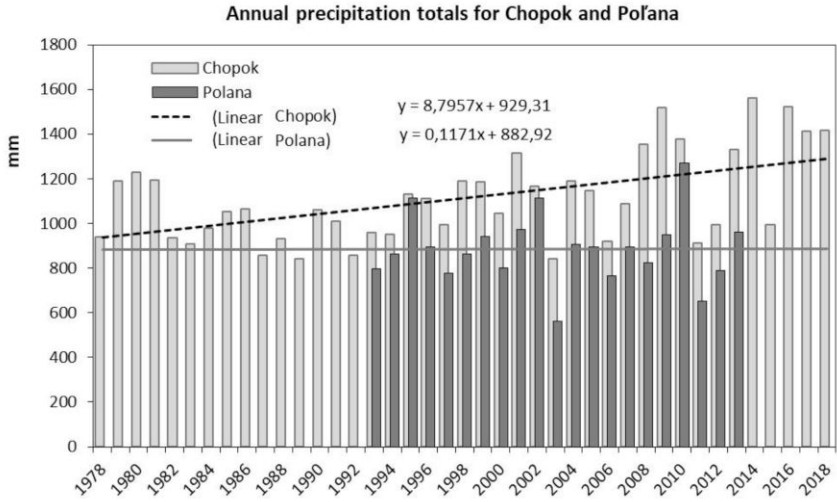

**Figure 3.** Annual precipitation totals for Chopok and Polana.

Total precipitation has been recorded in Poľana since 1993. During the analyzed 21-year period, precipitation totals did not show large fluctuations. The maximum total precipitation was recorded in 2010 at the level of 1270 mm. On average, precipitation at a given locality reaches 886 mm per year. Statistical analysis of precipitation total data obtained from this locality did not confirm the significance of the time trend. Using the Kruskal–Wallis test, we proved a statistically significant difference between these two localities in the years from 1993 to 2018 (see Table 1).

**Table 1.** Statistical differences between stations Chopok and Poľana based on concentrations of elements in precipitation.

| Kruskal–Wallis Test | | | Chopok–Poľana | |
|---|---|---|---|---|
| Component | Time Period | Count | *p*-Value | Significance |
| Precipitation | 1993–2013 | 21 | <0.0001 | *** |
| pH | 1995–2013 | 21 | 0.649 | NS |
| H+ | 1993–2013 | 21 | 0.2807 | NS |
| S-SO$_4$ | 1993–2013 | 21 | 0.3022 | NS |
| N-NO$_3$ | 1993–2013 | 21 | 0.252 | NS |
| N-NH$_4$ | 1995–2013 | 19 | 0.3205 | NS |
| Ca | 1993–2013 | 21 | 0.0019 | ** |
| Mg | 1995–2013 | 21 | <0.0001 | *** |
| K | 1995–2015 | 21 | 0.0056 | ** |

*** for $p < 0.001$, ** for $p < 0.01$, * for $p < 0.05$, NS for $p \geq 0.05$.

### 3.2. pH Values and H$^+$ Concentrations

Evaluation of temporal changes of pH values can be quantified only in the case of long-term stable observations performed at the same site. Figure 4 shows changes in pH and H$^+$ values during the observed period for Chopok 1978–2018 and Poľana 1993–2013. The Mann–Kendall test confirmed that the acidity of atmospheric precipitation in the observed time period shows a significant trend (see Table 1). At both studied localities, the pH value gradually increases slightly. As the pH increases, the concentration of hydrogen ions H$^+$ decreases naturally. This trend is mainly due to a decrease in the concentrations of major acidifying ions such as sulfate ions, nitrate ions. The decrease of these ions is evident from the following graphical representations (Figure 4).

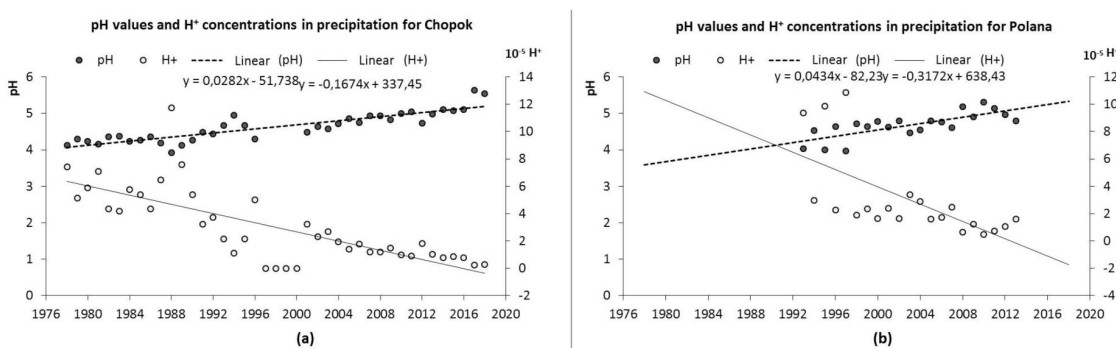

**Figure 4.** Annual mean weighted pH values and H$^+$ concentrations in precipitation: (**a**) Chopok, (**b**) Polana.

At Chopok, the pH value in annual averages ranges from 3.93 to 5.63. The lowest value is from 1988 and the highest from 2017. The recorded pH values of precipitation in Poľana reached their minimum of 3.96 in 1997 and the maximum of 5.4 in 1993. In Poľana the statistical significance of the increase in pH values is slightly lower than in Chopok, which can be caused by a shorter reference period. The given localities do not differ statistically significantly from each other in terms of comparison of pH values.

The time series and pH trend over a longer period indicate a decrease in acidity not only in Slovakia, but also in Europe [12,18,22]. The pH values correspond well with the pH values according to the EMEP maps (SHMÚ 2015). At the 22 sites within EMEP with long term pH measurements from 1980 to 2009 the average decrease in H+ concentration was 74% [5]. The increase of precipitation pH value in recent years in bulk precipitation and throughfall has been explained by a greater decrease in acidic anion concentrations [23].

The lowest pH in Europe is observed in the Eastern part of the continent which has relatively high sulfate deposition and a low base cation deposition [5]. Sicard et al. [24] published the pH values in precipitation with a significant decreasing trend of $-0.025 \pm 0.02$ unit pH year$^{-1}$ in France.

### 3.3. Sulfur and Nitrogen

A basic overview of the weighted annual concentrations of acidic elements (S-SO$_4$, N-NO$_3$, and N-NH$_4$) separately for the station Chopok and Poľana is shown in Figure 5. At both monitored stations, sulfate ions dominate in the rainwater. This is confirmed by work [25], that they obtained much higher mean values of sulfur deposition compared to the nitrate-nitrogen deposition—by more than 4-times in case of the open area in mountains in central part of Slovakia.

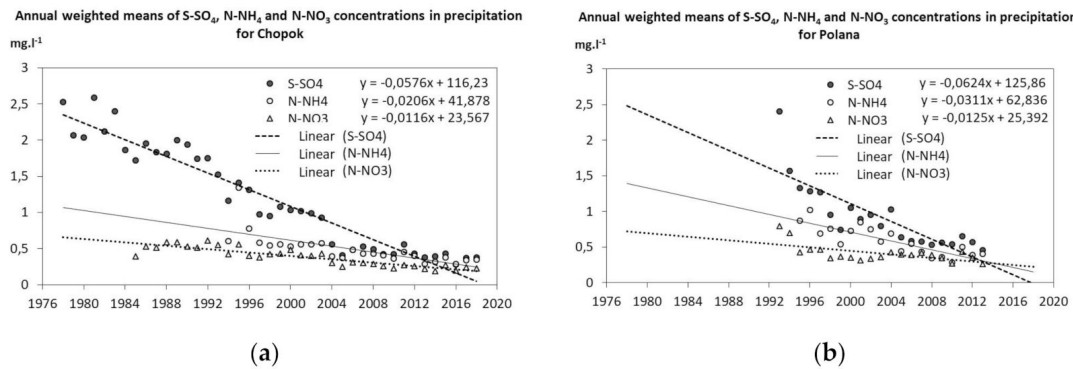

**Figure 5.** Annual mean weighted concentrations of acidic elements (S-SO$_4^{2-}$, N-NH$_4^+$, and N-NO$_3^-$) in precipitation: (**a**) Chopok, (**b**) Polana.

For the main air pollutants, the largest reductions across the EU-28 (in percentage terms) since 1990 have been achieved for SO$_x$ emissions (which decreased by 87%), followed by CO ($-66$%), NMVOCs ($-59$%), NO$_x$ ($-54$%), and NH$_3$ ($-27$%) [1].

Sulfate concentrations together with precipitation pH values have been monitored at the EMEP Chopok station for the longest of all monitored elements since 1978. At the Chopok station, the weighted annual mean sulfur concentrations represented the range 0.38–2.59 mg·L$^{-1}$ in the period 1978–2018. The highest value of the concentration of sulfur in sulfates in rainwater was recorded in 1981. The minimum value of the concentration was reached in 2016. In Poľana, the weighted annual averages of S-SO$_4$ concentrations ranged from 0.28 to 2.4 mg·l$^{-1}$. The values of sulfur concentration in SO$_4^{2-}$ in Poľana have been slightly higher in the last decade than the values at the Chopok station. The values of sulfate concentrations are characterized by considerable variability, and it is noticeable to see a gradual decrease in their dispersion. Over the last decade, the weighted annual average concentration of S-SO$_4$ has not exceeded 1 mg·l$^{-1}$.

Sulfate anions contribute the most to the acidity of precipitation, in Chopok for the whole monitored period from 1978–2018 their concentration decreased at a rate $-0.0576$ mg·l$^{-1}$ per year and in Poľana from 1993–2013 by $-0.0624$ mg·l$^{-1}$ annually. This decreasing trend of S-SO$_4$ concentration is significantly significant in both cases ($p < 0.0001$) (Table 2).

**Table 2.** Mann–Kendal test analyze and decrease of concentration elements in precipitation.

| | Variable | Time Period | Min | Max | Mean | Median | Std. Deviation | Sen's Slope | p-Value | Significance | Decrease % |
|---|---|---|---|---|---|---|---|---|---|---|---|
| Chopok | Precipitation | 1978–2018 | 840.5 | 1560.0 | 1118.4 | 1064.1 | 198.002 | 6.046 | 0.0079 | ** | −29.72 |
| | pH | 1978–2018 | 3.930 | 5.630 | 4.631 | 4.640 | 0.397 | 0.025 | < 0.0001 | *** | −28.77 |
| | $H^+$ | 1978–2018 | $2.34 \times 10^{-5}$ | $11.75 \times 10^{-5}$ | $3.32 \times 10^{-5}$ | $2.29 \times 10^{-5}$ | $2.58 \times 10^{-5}$ | 0.000 | < 0.0001 | *** | 92.99 |
| | $S\text{-}SO_4$ | 1978–2018 | 0.280 | 2.590 | 1.201 | 1.030 | 0.709 | −0.063 | < 0.0001 | *** | 84.49 |
| | $N\text{-}NO_3$ | 1985–2018 | 0.190 | 0.610 | 0.381 | 0.395 | 0.128 | −0.013 | < 0.0001 | *** | 53.15 |
| | $N\text{-}NH_4$ | 1994–2018 | 0.290 | 1.340 | 0.492 | 0.430 | 0.208 | −0.017 | < 0.0001 | *** | 63.97 |
| | Ca | 1992–2018 | 0.090 | 1.830 | 0.384 | 0.200 | 0.402 | −0.032 | 0.0001 | *** | 77.95 |
| | Mg | 1992–2018 | 0.020 | 0.330 | 0.062 | 0.040 | 0.068 | −0.005 | < 0.0001 | *** | 78.95 |
| | K | 1992–2018 | 0.040 | 0.700 | 0.156 | 0.090 | 0.147 | −0.011 | < 0.0001 | *** | 79.49 |
| Polana | Precipitation | 1993–2013 | 561.2 | 1270.4 | 886.0 | 892.7 | 152.4 | −0.009 | 0.0742 | NS | 13.35 |
| | pH | 1993–2013 | 3.960 | 5.310 | 4.680 | 4.720 | 0.350 | 0.023 | < 0.0001 | *** | −18.75 |
| | $H^+$ | 1993–2013 | $0.49 \times 10^{-5}$ | $10.87 \times 10^{-5}$ | $3.00 \times 10^{-5}$ | $1.91 \times 10^{-5}$ | $2.97 \times 10^{-5}$ | 0.000 | 0.0003 | *** | 84.83 |
| | $S\text{-}SO_4$ | 1993–2013 | 0.454 | 2.403 | 0.923 | 0.798 | 0.462 | −0.053 | < 0.0001 | *** | 68.45 |
| | $N\text{-}NO_3$ | 1993–2013 | 0.264 | 0.792 | 0.409 | 0.385 | 0.125 | −0.008 | 0.0134 | * | 45.01 |
| | $N\text{-}NH_4$ | 1995–2013 | 0.282 | 1.022 | 0.588 | 0.550 | 0.207 | −0.032 | < 0.0001 | *** | 49.99 |
| | Ca | 1993–2013 | 0.318 | 1.159 | 0.638 | 0.600 | 0.256 | −0.028 | 0.0011 | ** | 55.87 |
| | Mg | 1993–2013 | 0.050 | 0.295 | 0.135 | 0.124 | 0.064 | −0.008 | < 0.0001 | *** | 73.64 |
| | K | 1993–2013 | 0.066 | 1.223 | 0.410 | 0.252 | 0.332 | −0.015 | 0.0571 | NS | 57.75 |

*** for $p < 0.001$, ** for $p < 0.01$, * for $p < 0.05$, NS for $p \geq 0.05$; decrease was calculated from differences of the first three and last three year's averages.

The overall decrease in sulfate concentrations in the long-term time series corresponds to the decrease in $SO_2$ emissions since 1980 (SHMÚ 2018). From 1978 to 2018, sulfate concentrations in rainwater decreased at the Chopok station by 85.0% and at the Poľana station by 68.5% (1993–2013). This fact is in line with results from the monitoring made within EMEP. It shows large reductions in ambient concentrations and deposition of sulfur species during the last decades. Reductions are in the order of 70–90% since the year 1980 and correspond well with reported emission changes. As a result of the large reductions in sulfur concentrations, the acidity of precipitation has decreased across Europe (TØRSETH et al. 2012). According to [1] between 1990 and 2013, $SO_x$ emissions dropped in the Slovakia by 90% and in the EU-28 by 87% [1]. In 1990 the highest sulfur deposition areas were found in the central-east European areas in countries such as Germany, Poland, Czech Republic, and Slovakia. At year 2006 the highest load, although lower than previously, are found in eastern European countries such as Bulgaria, Romania, Serbia, and Bosnia and Herzegovina [4,26].

In France [24] the concentrations of $SO_4^{2-}$ and $nss\text{-}SO_4^{2-}$ (nss—non see salt) concentrations in precipitation have a significant decreasing trend, $-3.0 \pm 1.6$ and $-3.3 \pm 0.6\%$ year$^{-1}$, respectively, corresponding with the downward trends in $SO_2$ emissions in France ($-3.3\%$ year$^{-1}$). A good correlation ($R^2 = 0.84$) between $SO_2$ emissions and $nss\text{-}SO_4^{2-}$ concentrations was obtained. The decreasing trend of $NH_4^+$ was more significant ($-5.4 \pm 5.2\%$ year$^{-1}$) than that of $NO_3^-$ ($-1.3 \pm 2.4\%$ year$^{-1}$). In Latvia, the $SO_4\text{-}S$ ion concentrations in bulk precipitation also showed a significant negative linear trend [23].

Nitrates are involved in precipitation acidity to a lesser extent than sulfates [11,27]. With the measurement and determination of nitrate concentration in precipitation, Chopok began in October 1985 and Polana in 1993. The annual weighted means of nitrate concentrations converted to nitrogen at the Chopok precipitation ranged from 0.19 to 0.61 mg·l$^{-1}$. The minimum value is from 2014 and the maximum from 1992. In the monthly values, greater variability of nitrate concentrations was recorded. The N-NO3 concentration was 0.79 mg·l$^{-1}$ in the first year of measurement and the minimum value was 0.19 mg·l$^{-1}$ in the year 2014. The weighted annual concentrations of N-NO3 in Polana in the last decade exceeded the monitored values at Chopok.

The results of the analysis of time changes of nitrates concentrations in precipitation showed a statistically significant trend with decreasing tendency, as is evident from Figure 5 and Table 2. The rate of decrease in N-NO3 concentration in both stations is approximately the same, at Chopk is 0.0116 mg·l$^{-1}$ per year and at Poľana $-0.0125$ mg·l$^{-1}$ per year. The precipitation content of nitrates does not change as fast as the sulfate content.

At Chopok, nitrate concentrations decreased by 41.0% between 1985 and 2018 and 45.01% at Poľana for 1993–2013. Between 1990 and 2013, $NO_X$ emissions dropped in the Slovakia by 65% and in the EU-28 by 54% [1]. The reduction of $NO_X$ emissions in Europe from 1990 to 2009 were mainly caused by a change from burning of coal and gas to nuclear power. In Eastern Europe increased $NO_X$

emissions from road traffic after 2000. On the other hand, $NO_X$ emissions from traffic in Western European decreased, even though fuel consumption increased [5].

Monitoring of Ammonium ion in rainfall water Chopok started in 1994 and one year later at Poľana. At Chopok, the annual weighted mean of $N-NH_4$ concentrations ranged from 0.29 to 1.34 mg·l$^{-1}$ to the maximum value recorded in 1995. The maximum at Poľana was measured in 1996 with a value of 1.02 mg·l$^{-1}$ and a minimum in the year 2010 0.28 mg·l$^{-1}$. A statistically significant trend ($p < 0.0001$) (Table 2) with a decreasing trend for Chopok −0.0206 mg·l$^{-1}$ nitrogen per annum and Polana −0.0311 mg·l$^{-1}$ is observed from the analysis of time changes of ammonia nitrogen concentrations in precipitation at both stations.

### 3.4. Base Cations ($Mg^{2+}$, $Ca^{2+}$, $K^+$)

A basic overview of the weighted annual concentrations of base cations (Ca, Mg, and K) separately for the station Chopok and Poľana is shown in Figure 6. In Chopok, the measurement and determination of the concentration of basic cations in precipitation began in 1992, and a year later the measurement of the concentrations of these cations also began in Poľana.

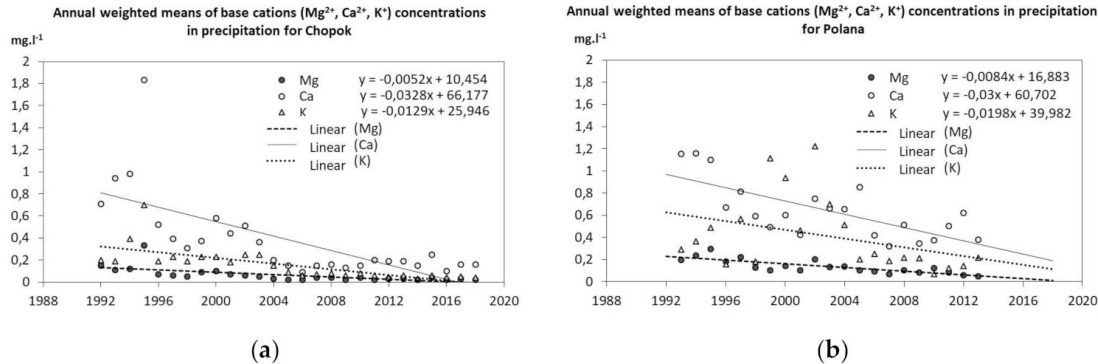

(**a**)　　　　　　　　　　　　　　　　　　　　　(**b**)

**Figure 6.** Annual mean weighted concentrations of base cations ($Mg^{2+}$, $Ca^{2+}$, $K^+$) in precipitation: (**a**) Chopok, (**b**) Polana.

At the Chopok station, for all the basic cations mentioned, the maximum value of the weighted annual concentration was recorded in 1995 for Ca 1.83 mg·l$^{-1}$, Mg 0.33 mg·l$^{-1}$, and K 0.7 mg·l$^{-1}$. A comparison of the concentration of basic cations in precipitation between stations shows mostly higher values at Poľana compared to EMEP station Chopok. In this case, since basic cations are the main component of terrestrial dust, the difference in the methods of measuring "wet-only" (rain gauges) vs. "bulk" (rainwater collectors) [28] is evident. In addition, basic cations are produced mainly in the production of building materials, as well as from local dust (roads, stone processing, etc.), which are more localized at lower altitudes [22,29].

From 1992 to 2018, there was an obvious decrease in basic cations in precipitation waters. A more significant decrease is at the Chopok station where the weighted annual average concentration of potassium decreased by 80.0%, magnesium by 86.7%, and calcium by 77.5%. In Poľana, within the concentration of the monitored basic cations, magnesium decreased the most by 73.6%, although the decrease of potassium by 57.8% and calcium by 55.9% is not negligible either.

Ca cations have the highest concentration from base cations observed in precipitation. The weighted annual means of calcium concentration in precipitation (Figure 6) at the Chopok station during the observed period varied in the range of 0.09–1.83 mg·l$^{-1}$ and at Poľana 0.318–1.159 mg·l$^{-1}$. The studied localities differ statistically significantly from each other (Table 2). Statistical analysis of the data confirmed a trend that is significantly significant (Table 1). In the case of both stations there is a decreasing tendency of calcium concentrations in precipitation, while at the station Chopok by −0.0328 mg·l$^{-1}$ per year and at Poľana by −0.03 mg·l$^{-1}$ per year.

A majority of the EMEP sites showed a decreasing trend of calcium in precipitation with an average decrease of 47% from 1980 to 2009 and 26% from 1990 to 2009. In the early 1990s, the closing of many lignite-fired power stations, iron, and steel smelters as well the implementation of effective abatement technologies for sulfur caused a reduction also in the emissions of base cations [5].

Magnesium concentrations are several times lower than calcium and potassium. When comparing the weighted annual averages of magnesium concentrations between the Poľana and Chopok stations in Figure 6. See the big differences among them. Precipitation from the Poľana station reaches significantly higher magnesium concentrations in almost all monitored years than at Chopok. We confirmed the statistically significant difference ($p < 0.0001$) between the given localities by the Kruskal–Wallis test. Concentrations in the range of 0.05–0.295 mg·l$^{-1}$ were recorded at Poľana and 0.02–0.33 mg·l$^{-1}$ at Chopok. At both stations we can confirm a statistically significant trend ($p < 0.0001$) with a decreasing tendency, in the case of Poľana it is a decrease of −0.0084 mg·l$^{-1}$ and at the station Chopok −0.0052 mg·l$^{-1}$.

At Chopok station, potassium concentrations change only at a very narrow interval. With regard to the values determined from the precipitation at Poľana, this statement is valid only in the last evaluated years. Values in Poľana stabilized relatively after 2005. Weighted annual potassium concentrations ranged from 0.03 to 0.7 mg·l$^{-1}$ for Chopok and 0.07 to 1.22 mg·l$^{-1}$ for Poľana throughout the period under review. The Mann–Kendal test confirmed a statistically significant trend with a declining character at the Chopok station. In the long run, the trend shows a decrease in potassium concentration by −0.0129 mg·l$^{-1}$ per year. The weighted annual averages of K concentrations in Poľana did not show a statistically significant trend. From Figure 6, however, it can be seen that the concentrations of K reach lower values than in the first half of the observed period. Using the Kruskal–Wallis test, we proved a statistically significant difference between the Chopok and Poľana stations.

## 4. Discussion

The long-term changes of chemical composition of atmospheric precipitation depends on many factors, but the decisive factors are changes in air pollutant emissions as well as changes in meteorological processes affecting their transformation into a liquid phase in clouds and precipitation [30,31]. The resulting changes are thus an integrated indicator of complex physico-chemical transformations of pollutants from the emission source to the captured precipitation at a specific location [30,32–34]. The results of many observations, especially in the northern hemisphere, point to significant changes in precipitation chemistry, especially in pH values, concentrations of sulfur, nitrogen, but also basic cations. Most notably, these changes are visible at pH values and sulfate concentrations in precipitation water [2,11,24,27,34,35].

Several studies confirm that the primary cause of the decrease of pollutants in atmospheric precipitation is a decrease in sulfur and nitrogen emissions due to the application of measures under the Convention on Long-Range Transboundary Air Pollution—CLRTAP [1,2,7], although the decrease in emissions does not fully explain all observed pollutant trends [3]. Changes in other factors such as changes in air temperature and precipitation, changes in atmospheric circulation, or the occurrence of extreme weather events must also be taken into account [30,32,33].

Trends in decreasing sulfur and nitrogen concentrations in precipitation over the last 2–3 decades have been recorded mainly in Europe and North America, although the intensity of the decrease has been regionally different [2,4,5,23,27,34]. The most significant changes have been identified in Central and Eastern Europe, mainly due to structural changes in energy and industry sectors [1,2,4,5]. Recently studies have emerged on the possible impact of climate change on the further development of acidic and basic components contained in atmospheric precipitation, in particular the impact of changes in precipitation (decrease or increase in precipitation totals), air temperature and atmospheric circulation changes on long-range transmission, and chemical transformation of pollutants in gas and liquid phases [11,12,32,33]. For this reason, it is therefore necessary to maintain the monitoring of rainwater quality as much as possible, especially in localities with their long-term measurement.

Changes in the chemical composition of precipitation also significantly affect the biogeochemical cycles in forest and aquatic ecosystems and can contribute to changes in habitat diversity and biocenoses as well as changes in biomass production (e.g., changes in nitrogen content in atmospheric precipitation). In areas with sufficient precipitation, long-term changes in the chemical composition of precipitation will gradually be reflected in the chemical regime of surface and groundwater [35–37]. The issue of the economic effects of air pollution and precipitation quality, which may take on a different dimension in climate change, must not be forgotten either [38].

## 5. Conclusions

The presented work deals with the issue of long-term changes in the quality of precipitation in mountain areas of Slovakia at the station EMEP Chopok and VDO Poľana-Hukavský grúň. A 41-year time series of measured data of most of the analyzed elements of precipitation chemistry was available for the Chopok locality. At the Poľana station, the concentrations of elements in precipitation are monitored and are available for a 21-year period. In this work we evaluated the long-term development of concentrations of elements in precipitation ($S-SO_4$, $N-NO_3$, $N-NH_4$, Ca, and Mg, K).

The concentration of sulfur in sulfates in precipitation water decreases significantly at both monitoring stations. At Chopok the decrease represents up to 84.5% decrease compared to the first three years of measurement and at Poľana 68.5% decrease. Nitrogen concentrations in nitrate and ammonium ions decreased with a significant trend at both monitored mountain stations. In both cases, its deposition decreased by almost 50%. Nitrogen concentrations in both nitrate and ammonium ions decreased at approximately the same rate per year at both stations.

For all basic cations, a decrease in their concentration in atmospheric precipitation is observed, with the exception of the potassium cation at the Poľana station, where the trend was not statistically confirmed.

**Author Contributions:** Conceptualization, J.M. and J.Š.(Jaroslav Škvarenina); methodology, M.H. and J.M.; software and statistics, J.Ď. and M.H.; data management, S.T.; writing—original draft preparation, J.M. and M.H.; writing—review and editing, J.Š.(Jana Škvareninová). All authors have read and agreed to the published version of the manuscript.

**Funding:** This research received no external funding.

**Acknowledgments:** This work was accomplished as a part of VEGA projects No.: 1/0500/19, 1/0111/18 of the Ministry of Education, Science, Research, and Sport of the Slovak Republic and the Slovak Academy of Science; and the projects of the Slovak Research and Development Agency No.: APVV-18-0347; APVV-15-0425 and APVV-19-0340. The authors thank the agencies for the support.

**Conflicts of Interest:** The authors declare no conflict of interest.

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
