# Peer review of "Long-Term Temporal Changes of Precipitation Quality in Slovak Mountain Forests"

_water, doi:10.3390/w12102920_

Round 1
Reviewer 1 Report
The present paper brings the analysis of the long term monitoring of the precipitation in the Slovakia both form the quantitative and qualitative points of view. The authors used the data sets from two stations located in the various altitude and described the variability of the amount and quality of the rainfall water. I found out of the manuscript as interesting, valuable and suitable for publishing.
I have a few questions and/or recommendations:
- Although I am certainly not fully qualified to judge the English, I feel that the authors used rather simple English and that is why I strongly recommend to improve language and style by native speaker.
- Lines 23 and 55: I suppose that the correct form is “precipitation” (singular).
- Lines 78 and 86: I suggest: … mean annual precipitation total…
- Line 106: I recommend to reformulate the sentence. Due to my opinion that is only the recalculation according to the atomic weights.
- Line 126: It rather curious to state the precipitation totals in the form of two decimal places.
- Please correct Figs. 3, 4, 5 and 6 – the legends are in Slovak.
- I would suggest to add also the medians into the Table 1.
- Lines 253 and 254: What kind of the collector was used in fact? I should be mentioned in the chapter 2. Materials and Methods.
- Lines 260 and 261: It is needless to state the percentage in the form of two decimal places.
- Line 417: The correct name of one co-author is Hejzlar.
Author Response
Although I am certainly not fully qualified to judge the English, I feel that the authors used rather simple English and that is why I strongly recommend to improve language and style by native speaker.
The proof of English will be performed after professional revision process.
Lines 23 and 55: I suppose that the correct form is “precipitation” (singular).
Accepted and corrected.
Lines 78 and 86: I suggest: … mean annual precipitation total…
Accepted and corrected.
Line 106: I recommend to reformulate the sentence. Due to my opinion that is only the recalculation according to the atomic weights.
Accepted and corrected.
Line 126: It rather curious to state the precipitation totals in the form of two decimal places.
Accepted and corrected.
Please correct Figs. 3, 4, 5 and 6 – the legends are in Slovak.
Accepted and corrected.
I would suggest to add also the medians into the Table 1.
Accepted, the median values have been added into Table1
Lines 253 and 254: What kind of the collector was used in fact? I should be mentioned in the chapter 2. Materials and Methods.
Accepted, information related to collectors has been added into methods.
Lines 260 and 261: It is needless to state the percentage in the form of two decimal places.
Accepted and corrected.
Line 417: The correct name of one co-author is Hejzlar.
Accepted and corrected.

Reviewer 2 Report
This manuscript reports the result of a time series analysis of precipitation quality in Slovak forests. Authors used archived data from 2 stations in the study area. In general, the manuscript is very short and has a lack of information regarding the methodology section. Authors did not provide any information for their analysis and the studied quality parameters. Moreover, the manuscript organization and English language need much efforts before publication. However, I have the following specific comments:
L46: You did not define this term before EMEP. Please define it.
The introduction section is too short and the authors did not report their study objectives.
L74: Please define the term SHMI.
Figure 1: Add the north arrow to the map.
L95: You did not provide any information about the studied precipitation quality parameters. This section needs more details and description on how you performed the analysis?
L112-116: Move this part to your introduction section.
Figure 3: Polana data stopped from 2014, Why? You did not mention anything about this. Also, I don't see any need for the reported equations in this graph.
Section 3.2, 3.3 & 3.4: You did not report any details about these measurements through your methodologies. Please describe your methodology.
L194-217: Most of the information that you reported here are not your results. This part could be used in your discussion.
L244: Section 3.4. Modify the section number.
Author Response
The proof of English will be performed after professional revision process.
L46: You did not define this term before EMEP. Please define it.
Accepted and corrected.
The introduction section is too short and the authors did not report their study objectives.
Accepted. The study objectives have been added into Introduction section.
L74: Please define the term SHMI.
Accepted and corrected.
Figure 1: Add the north arrow to the map.
Accepted and corrected.
L95: You did not provide any information about the studied precipitation quality parameters. This section needs more details and description on how you performed the analysis?
Accepted, information related to analysis has been added into methodology.
L112-116: Move this part to your introduction section.
Noted.
Figure 3: Polana data stopped from 2014, Why? You did not mention anything about this. Also, I don't see any need for the reported equations in this graph.
The reasons for Polana data stopped were added into methodology. The reported equations in graphs we consider to be important due to two reasons:
- From the equation I can obtain information about the mean annual increase/decrease of the parameter
- Equation in graph can help to compare our results with other long-term precipitation quality measutrements
Section 3.2, 3.3 & 3.4: You did not report any details about these measurements through your methodologies. Please describe your methodology.
Accepted, Methodology description was reedited.
L194-217: Most of the information that you reported here are not your results. This part could be used in your discussion.
The redaction recommendations for the discussions did not allow to describe the problems of the authorization of the data and results.
May be some clarification: J. Mindas has established the Research Demonstration Object at Polana (as a scientist from Forest Research Institute in Zvolen), and since 1992 he performed the precipitation measurements, water sampling and chemical analyses under the ICP Forests monitoring programme. He was responsible for deposition monitoring up to 2004, the continuation of activities at Polana have been performed by S. Tóthová (up to 2019). It means that both persons are fully responsible for the chemical data for Polana.
Chopok data are operated by the Slovak Hydrometeorological Institute (SHMI) in Bratislava, and these data are opened through the EMEP Program. J.Skvarenina and J.Mindas were the partners into many air pollution projects together with D.Zavodsky and M.Mitosinkova (responsible persons for EMEP data from SHMI, Mr. Zavodsky died, Mrs. Mitosinkova is now retired). The Chopok data we used in many other projects and papers, and have been used as comparable data for Polana data evaluation.
L244: Section 3.4. Modify the section number.
Accepted and corrected.

Reviewer 3 Report
The article entitled “Long-term temporal changes of precipitation quality in Slovak mountain forests”, a very nice temporal investigation of the precipitation quality in Slovak forest.
The topic is really interesting for the scientific community and the data are valuable. The article is very nice due to describing the monitoring of not one but two monitoring stations at different altitudes.
The paper is well written (for no mother tong english speaker); I could follow the thread of the discussion. In the introduction, the scientific content and state of the art are clearly written. The methodology and the data are clearly discussed and also described from a statistically point of view.
Author Response
The article entitled “Long-term temporal changes of precipitation quality in Slovak mountain forests”, a very nice temporal investigation of the precipitation quality in Slovak forest.
Noted.
The topic is really interesting for the scientific community and the data are valuable. The article is very nice due to describing the monitoring of not one but two monitoring stations at different altitudes.
Noted.
The paper is well written (for no mother tong english speaker); I could follow the thread of the discussion. In the introduction, the scientific content and state of the art are clearly written. The methodology and the data are clearly discussed and also described from a statistically point of view.
Noted.

Round 2
Reviewer 2 Report
Authors improved their manuscript while I still believe that the introduction section is not describing the research problem. Please elaborate and improve this section. Moreover, the English revision is needed before publication.
Figure 1: Adjust the figure as it has an outlier section (left part).
Author Response
Thank you for the second round of revision.
- The introduction part has been changed, new additional information has been added.
- Mentioned figure has been adjusted.
Regards
Jozef Mindas
